# Proteomic Profiling of Inflammatory Protein Dysregulation in HLA-B27-Positive Ankylosing Spondylitis: Molecular Signatures and Potential Biomarkers

**DOI:** 10.3390/biom15040516

**Published:** 2025-04-01

**Authors:** Yuzhu Yan, Jihan Wang, Yangyang Wang, Junye Liu, Wenjuan Yang, Min Niu, Yan Yu, Heping Zhao

**Affiliations:** 1Clinical Laboratory of Honghui Hospital, Xi’an Jiaotong University, Xi’an 710054, China; 2Shaanxi Provincial Key Laboratory of Infection and Immune Diseases, Shaanxi Provincial People’s Hospital, Xi’an 710068, China; 3School of Electronics and Information, Northwestern Polytechnical University, Xi’an 710129, China; 4Department of Rheumatology Immunology and Endocrinology, Honghui Hospital, Xi’an Jiaotong University, Xi’an 710054, China

**Keywords:** ankylosing spondylitis, proteomics, HLA-B27, inflammatory biomarkers, cytokine dysregulation

## Abstract

This study explored the proteomic landscape of inflammatory protein dysregulation in ankylosing spondylitis (AS), a chronic inflammatory disorder primarily affecting the axial skeleton and strongly associated with the HLA-B27 allele, particularly the HLA-B2705 and HLA-B2704 subtypes prevalent in Chinese populations. Blood samples from HLA-B27-positive AS patients and normal controls (NC) were analyzed using the Olink Target 96 inflammation panel to profile 92 inflammatory proteins. HLA-B27 subtyping was performed via PCR-SSP. To identify key proteins and stratify AS subtypes, we employed machine learning classifiers, including LightGBM models coupled with SHAP value interpretation, alongside traditional statistical analyses. The proteomic analysis revealed significant dysregulation of pro-inflammatory cytokines, such as IL-6 and IL-17A, in AS patients compared to NC, with CXCL9 and NRTN identified as potential biomarkers associated with disease activity. The combination of LightGBM classifiers and traditional statistical methods demonstrated high accuracy in distinguishing AS from NC and effectively stratifying subtypes. These findings provide valuable insights into the inflammatory mechanisms underlying AS pathogenesis and highlight potential biomarkers and therapeutic targets for improving diagnosis and treatment strategies. Future studies with larger and more diverse cohorts, as well as longitudinal designs, are warranted to validate these biomarkers and elucidate their dynamic changes during disease progression.

## 1. Introduction

Ankylosing spondylitis (AS; also known as radiographic axial spondyloarthritis, r-axSpA) is a common chronic inflammatory rheumatic disease that primarily affects the axial spine and sacroiliac joints, leading to chronic back pain, spinal stiffness, and impaired mobility [1,2,3,4]. The diagnostic prevalence of AS in the United States is reported to be 0.09% [5], while global estimates range from 0.1% to 1.4%, with significant geographic variations [6,7]. Systematic reviews have found the highest prevalence in North America, followed by Europe, Asia, Latin America, and Africa [6]. Notably, Asian populations, including China, have shown varying prevalence rates, reflecting the broader global trends [8]. AS is nearly twice as common in men as in women, a notable exception given that autoimmune conditions are generally more prevalent among females [5,9].

A significant breakthrough in understanding AS came with the discovery of the association between the human leukocyte antigen-B*27 (HLA-B27) antigen and the disease, yet the mechanistic link between HLA-B27 and AS remains one of the great enigmas in rheumatology [10]. This association, discovered by Schlosstein et al. and Brewerton et al. in 1973 over 50 years ago, represents the strongest known association of a common variant with any human disease [11,12]. The genetic heterogeneity of HLA-B27 is well-documented, with numerous subtypes identified to date. The most common subtypes associated with AS are HLA-B2705 in Caucasians, HLA-B2704 in Chinese, and HLA-B2702 in Mediterranean populations. The prevalence of AS in China ranges from 0.20% to 0.42%, with 88.8% to 89.4% of AS patients testing positive for HLA-B27. Among Chinese AS patients, HLA-B2704 is the most common subtype, followed by HLA-B2705 [8]. HLA-B27:04 is associated with a higher risk of AS in Chinese populations compared to HLA-B27:05 and is strongly linked to AS in Asian populations overall [13,14]. The differential distribution and potential functional disparities between these subtypes may contribute to the variable clinical phenotypes observed in patients.

Currently, there is no cure for AS in clinical practice. The primary goals of AS treatment are to control inflammation, alleviate symptoms, prevent deformities, and thereby improve the quality of life. Systemic inflammation, driven by a dysregulated immune response, is a key pathological hallmark of AS, highlighting the importance of investigating inflammatory markers. Given the central role of inflammation in disease progression, identifying specific inflammatory mediators in AS could aid in the discovery of reliable diagnostic biomarkers and potential therapeutic targets. While previous studies have focused on genetic and transcriptomic profiling, comprehensive proteomic analyses—particularly those examining inflammatory protein expression—remain relatively limited, underscoring the need for further investigation in this area. For instance, studies have identified and validated related gene expression signatures in AS patients, highlighting key genes such as *SPOCK2* and *EP300* that are involved in inflammation and joint destruction, potentially driving disease progression [15]. Additionally, transcriptomic analyses have uncovered distinct gender-specific patterns in Th17-related inflammatory pathways in AS, with males exhibiting upregulation of key pro-inflammatory genes, which may contribute to gender differences in disease expression [16]. Furthermore, a DNA methylation study has revealed dysregulation of *LGR6* DNA methylation and reduced transcript levels in AS patients, suggesting a significant epigenetic link to AS pathogenesis and its inflammatory processes [17]. Despite significant advances in understanding the genetic expression patterns and epigenetic abnormalities in AS patients, there remains a critical need for more in-depth studies focused on proteomics. Proteomic studies are pivotal in linking genetic and epigenetic alterations to functional protein-level changes, thereby providing a more comprehensive understanding of disease mechanisms. Proteins, being the final products of gene expression, directly influence various biological processes, including inflammation and disease progression [18,19].

To address this gap, we selected the Olink Target 96 inflammation panel for this study, a highly sensitive and multiplexed proteomic assay that enables the quantification of 92 inflammation-related proteins with high specificity and reproducibility. This technology is based on a proximity extension assay (PEA), which combines antibody-mediated recognition with DNA polymerase-driven amplification, allowing for precise and high-throughput protein quantification [20,21,22,23]. Compared to conventional proteomic methods such as enzyme-linked immunosorbent assay (ELISA) and mass spectrometry-based approaches, the Olink PEA offers a broader dynamic range, reduced sample consumption, and improved detection sensitivity, making it particularly well-suited for analyzing low-abundance inflammatory biomarkers in clinical samples. The selection of this panel was driven by its ability to capture a comprehensive spectrum of inflammatory proteins relevant to AS, providing insights into cytokine-mediated immune dysregulation and potential disease-specific biomarker signatures. In this research, we explore the immunological landscape of AS serum samples using the Olink Target 96 inflammation panel. By focusing on patients with distinct HLA-B27 subtypes, we aim to identify specific inflammatory signatures that may correlate with disease phenotypes and susceptibility. Additionally, this investigation integrates molecular-level data to shed light on the biomolecular underpinnings of AS, to elucidate disease mechanisms and identify potential therapeutic targets. Moreover, this study’s findings provide preliminary insights that may help improve our understanding of the associations between HLA-B27 and spondyloarthropathies.

## 2. Materials and Methods

### 2.1. Study Design

The experimental design and research workflow of this study are illustrated in Figure 1. The process mainly contains participant recruitment, data collection, sample processing, a protein analysis, and a statistical analysis, providing a detailed investigation of the inflammatory protein landscape in HLA-B27-positive AS patients.

### 2.2. Participant Recruitment

This cross-sectional study was approved by the Biomedical Research Ethics Committee of Hong Hui Hospital, Xi’an Jiaotong University (No. 202312027). Written informed consent in the local language was obtained from all participants at enrollment.

The cohort of HLA-B27-positive AS patients was recruited from the Department of Rheumatology Immunology and Endocrinology of the participating hospital between January 2023 to December 2023. The eligible participants were adults diagnosed with AS according to the modified New York criteria [24]. The diagnostic criteria for AS include both clinical and radiological features. Clinically, patients exhibit persistent lower back pain for at least 3 months that improves with activity but not with rest, limited movement in the lumbar spine in both the sagittal and frontal planes, and a chest expansion measurement below the normal range. Radiologically, the presence of bilateral sacroiliitis of grade II–IV or unilateral sacroiliitis of grade III–IV on an X-ray is required. A diagnosis of AS can be made if the radiological criterion is met along with at least one of the clinical criteria. The exclusion criteria for this study included complete spinal ankylosis; the presence of other systemic inflammatory or chronic pain diseases; active inflammatory diseases such as Crohn’s or ulcerative colitis; recent cardiovascular events; a history of malignancy or lymphoproliferative disorders; major surgery within 6 months; MRI contraindications; unstable organ diseases; recent use of biologics, live vaccines, high-dose corticosteroids, potent opioids, or disease-modifying antirheumatic drugs (DMARDs); participation in other clinical trials; active or latent tuberculosis; and breastfeeding women. The NC participants were recruited from patients who received treatment or underwent physical examinations at the participating hospital. To minimize potential bias, stringent inclusion criteria were applied to the NC group. Participants with any comorbidities known to affect their immune status or inflammatory markers were excluded to ensure the NC group represented a healthy population without confounding factors that could influence the inflammatory profiles.

### 2.3. Data Collection

Demographic and clinical data were collected for each participant, including their age, gender, weight, disease duration, and medication interventions. Their disease activity and functional status were assessed using the BASDAI and the BASFI, respectively [25].

### 2.4. Sample Collection and Processing

Peripheral blood samples were collected in blood collection tubes from all participants in the clinical laboratory of the participating hospital. For serum Olink proteome detection, the blood samples were centrifuged at 1000–2000× *g* for 10 min at 2–8 °C to remove clots. The resulting serum was promptly transferred to clean tubes and stored at −80 °C until a further analysis.

### 2.5. Clinical Laboratory Test

The clinical laboratory tests for the WBC count, CRP levels, and ESR were conducted using the Sysmex XN-9100 hematology analyzer (Sysmex, Kobe, Japan), Roche Cobas 8000 automatic biochemical analyzer (Roche, Basel, Switzerland), and Succeeder SD-1000 dynamic ESR analyzer (Beijing Succeeder Technology Inc., Beijing, China), respectively.

### 2.6. HLA-B27 Subtyping

Genomic DNA was extracted from blood samples collected from AS patients and NC subjects using a standard DNA extraction kit (Xi’an Tianlong Science and Technology Co., Ltd., Xi’an, China). HLA-B27 genotyping and subtype determination were performed using the polymerase chain reaction with sequence-specific primers (PCR-SSP) method, according to the manufacturer’s instructions provided by Xi’an Tianlong Science and Technology Co., Ltd. The PCR products were analyzed using agarose gel electrophoresis, and the subtypes were identified based on the presence of specific bands corresponding to each HLA-B27 subtype.

### 2.7. Inflammatory Protein Analysis

Inflammatory proteins in serum samples from AS patients and NC participants were analyzed using the Olink Target 96 inflammatory panel (Olink Proteomics AB, Uppsala, Sweden). This method employs PEA technology [26], enabling the simultaneous analysis of 92 inflammatory proteins from just 1 µL of serum. The process involves pairs of oligonucleotide-labeled antibody probes binding to targeted proteins. When these probes are in close proximity, a unique DNA sequence is generated through a proximity-dependent DNA polymerization event. This sequence is then detected and quantified using a microfluidic real-time PCR instrument (Signature Q100, Olink Proteomics, Uppsala, Sweden). The raw Ct data were subjected to a two-step quality control (QC) and normalization process using both internal and external controls, as recommended by the manufacturer. Specifically, four internal controls were added to each sample to monitor the assay performance and sample quality. First, each sample plate was evaluated based on the standard deviation of the internal controls, which was required to be below 0.2 NPX; only data from plates that met this criterion were included in the analysis. Second, the quality of individual samples was assessed by measuring the deviation from the median value of the controls, with samples deviating by less than 0.3 NPX considered to have passed the QC, while those failing this threshold were flagged in the output file under the “QC warning” column and interpreted with caution. After passing QC, the Ct data were normalized and converted to normalized protein expression (NPX) values on a log2 scale, providing a relative quantification of the proteins in the samples. A higher NPX value indicates greater protein expression. This method ensures high specificity and sensitivity, making it suitable for detecting potential serum or plasma biomarkers of inflammation in AS and NC participants.

### 2.8. Statistical Analysis

Statistical analyses were performed mainly using R (Version 4.3.1) for the data processing, analysis, and visualization. The Wilcoxon rank-sum test was employed to identify differential proteins between AS patients and NC subjects, with *p*-values < 0.05 considered as significant differences. A principal component analysis (PCA) was performed using the “FactoMineR” and “factoextra” packages to visualize the separation of inflammatory protein profiles between groups. The LightGBM (light gradient boosting machine) classifier, along with SHAP (Shapley Additive Explanations) plots generated by the “lightgbm” and “shapviz” packages, was used to identify the most discriminative proteins between groups. Receiver operating characteristic (ROC) curves and a regression analysis with MedCalc statistical software (Version 19.0.4) were used to assess the diagnostic accuracy levels of the selected proteins between groups. A correlation analysis was conducted using the “PerformanceAnalytics” package to evaluate the relationships among the clinical parameters of AS patients in Table 1. A Spearman correlation analysis was performed to analyze the correlations between protein expression and BASDAI and BASFI scores, with *p*-values < 0.1 considered as significant differences.

## 3. Results

### 3.1. Clinical Characteristics of the Participants

The study successfully recruited a total of 53 participants, comprising 43 AS patients and 10 normal control (NC) participants, based on predefined inclusion and exclusion criteria. Detailed clinical characteristics for all participants are presented in Table 1. According to the study design, all AS patients were HLA-B27-positive, resulting in a 100% HLA-B27 positivity rate within the AS group. The clinical information revealed no significant differences between AS patients and NC controls in terms of age distribution, gender composition, and body mass index (BMI). Both groups had a comparable mean age and a balanced gender ratio, ensuring reliable comparisons. Additionally, the white blood cell (WBC) counts were similar across both groups, with the majority falling within the normal reference range (Table 1 and Figure 2A). The comparability in the basic clinical information and laboratory results underscores the suitability of both the AS patients and NC controls for subsequent analyses in this study. Notably, the AS group exhibited higher levels and abnormal rates of C-reactive protein (CRP) and erythrocyte sedimentation rate (ESR) scores compared to the NC group, where almost all CRP and ESR levels were within normal ranges (Table 1 and Figure 2A). This indicates that the AS group had a higher incidence of elevated CRP and ESR scores, reflecting increased inflammation in these patients (Figure 2B). We subsequently performed a correlation analysis of the clinical parameters among the 43 cases of AS patients. The results illustrated that there is a significant positive correlation between CRP levels and the ESR, indicating that both markers are reflective of the inflammatory status in AS patients. Additionally, the Bath Ankylosing Spondylitis Disease Activity Index (BASDAI) and the Bath Ankylosing Spondylitis Functional Index (BASFI) scores showed a positive correlation, suggesting that higher disease activity is associated with greater functional impairment in AS patients.

### 3.2. Analysis of Inflammatory Protein Profiles in AS Patients and NC Subjects

The Olink Target 96 inflammation panel used in this study measures 92 protein biomarkers per sample. During data preprocessing, proteins with missing values in more than 50% of the samples were excluded, resulting in a final set of 81 proteins for the subsequent analysis. Detailed information and expression profiles of these 81 proteins are provided in Appendix A. A PCA of the entire set of 81 proteins was initially performed to obtain an overview of the inflammatory protein profiles between AS patients and NC subjects, as shown in Figure 3. While the PCA plot (Figure 3A) did not reveal a very distinct separation between the AS patients and NC controls, it provided a preliminary indication of potential differences. Similarly, the PCA plot (Figure 3B) showed no obvious clustering patterns between AS subtypes based on the whole protein profile. These results underscored the need for a further analysis focusing on a more selective set of biomarkers. Since the majority of the 43 AS patient samples were classified as either HLA-B2704 or HLA-B2705, with only four samples belonging to other subtypes, the subsequent analyses focused solely on the HLA-B2704 and HLA-B2705 subtypes. Other HLA-B27 subtypes were excluded from the further detailed analysis.

### 3.3. Identification of Protein Biomarkers Differentiating AS Patients from NC Subjects

To identify protein biomarkers that differentiate AS patients from NC subjects, we initially used the Wilcoxon rank-sum test to detect differences in protein levels between the two groups. Proteins with a *p*-value < 0.05 were considered statistically significant between the comparison groups. As shown in Table 2, there are four, six, and six significantly differentially expressed proteins among the three comparison groups. The analysis revealed an upregulation of pro-inflammatory cytokines such as IL-6 and IL-17A in AS patients compared to NC subjects (*p*-value < 0.05). Certain pro-inflammatory cytokines, including TNF and IFN-γ, exhibited elevated levels in AS patients, whereas the anti-inflammatory protein IL-10 was downregulated, although these differences were not statistically significant when compared to NC subjects (Appendix A). This elevation suggests an active inflammatory response characteristic of AS patients.

Next, we utilized the LightGBM classifier to identify the top ten proteins that effectively differentiate between the various groups. The combination of the LightGBM model and SHAP visualizations allowed us to pinpoint the most influential proteins for classification. As illustrated in the SHAP plots (Figure 4), these proteins played a crucial role in distinguishing AS (total) from NC, AS_HLA-B2704 from NC, and AS_HLA-B2705 from NC. Specifically, IL-6, CCL11, and CCL25 were identified by both the Wilcoxon test and the LightGBM classifier for distinguishing AS (Total) from NC. IL-6, IL-17A, and ST1A1 were identified as overlapping biomarkers for differentiating AS_HLA-B2704 from NC, while CCL25, LIF-R, and NRTN were identified for distinguishing AS_HLA-B2705 from NC (Figure 5). The intersection of proteins identified by both methods was further analyzed using an ROC analysis and PCA. Notably, all of these biomarkers demonstrated good diagnostic performance with area under curve (AUC) values greater than 0.7 in the ROC analysis. Moreover, compared to the initial PCA plot based on all 81 proteins profiled in Figure 2, the PCA using these filtered biomarkers revealed a much clearer separation between each pair of comparison groups, underscoring the improved discriminative power of the selected markers (Figure 5). These findings highlight the potential of the identified proteins as reliable biomarkers for distinguishing AS patients from NC subjects, as well as for differentiating among various AS subtypes.

### 3.4. Correlation Between Protein Expression and BASDAI and BASFI Scores in AS Patients

The BASDAI and BASFI scores are clinical parameters used to assess disease activity and functional impairment, respectively, in AS patients. We performed a Spearman correlation analysis to explore the relationship between protein expression profiles and BASDAI and BASFI scores. Our analysis revealed a positive correlation between the expression levels of certain inflammatory proteins and BASDAI scores. For example, as shown in Figure 6A, CXCL9 expression exhibited a positive correlation coefficient of 0.3849 with BASDAI scores. Similarly, NRTN expression showed a Spearman correlation coefficient of 0.2588 (Figure 6B). These correlations highlight the potential of these proteins as biomarkers for disease activity and could aid in developing personalized treatment strategies. In contrast, no statistically significant correlations were found between protein expression and BASFI scores, suggesting that the inflammatory proteins measured may be more indicative of disease activity rather than functional impairment in AS patients.

## 4. Discussion

According to the new nomenclature, AS is referred to as r-axSpA. In this study, we retained the term AS for consistency with the majority of cited references. This study provides a comprehensive analysis of the inflammatory protein landscape in HLA-B27-positive AS patients, offering valuable insights into disease-associated inflammatory processes. To the best of our knowledge, this is the first study to employ the Olink Target 96 inflammation panel for a high-throughput and sensitive assessment of inflammatory proteins in AS. This advanced proteomic approach enabled the identification of subtle yet significant differences in protein expression between AS patients and healthy controls, as well as among different AS subtypes.

The findings highlight the dysregulation of key inflammatory proteins, reinforcing their relevance to AS pathogenesis. Pro-inflammatory cytokines, including IL-6, IL-17A, TNF, and IFN-γ, play central roles in AS by promoting inflammation and immune dysregulation [27,28]. IL-6 and IL-17A, in particular, have been implicated in previous studies, where elevated levels were associated with disease activity and treatment response [29]. A related study found that serum IL-6 and TNF-α levels were significantly higher in AS patients compared to controls, with even higher levels in patients during active disease phases. Both IL-6 and TNF-α levels were positively correlated with CRP and ESR levels, indicating their involvement in the inflammatory response and disease activity of AS [30]. Researchers identified distinct gene expression patterns in IL-17-enriched peripheral blood mononuclear cells (PBMCs) between male and female AS patients. Males showed upregulation of TGF-β, PGE2, S100 proteins, and IL-17RC and downregulation of Th17-associated receptors and specific Th17 differentiation genes compared to females. These differences in pro-inflammatory gene expression may underlie the gender-specific phenotypic variations observed in AS [16]. Additionally, elevated blood type I and II IFN activity and serum levels of IFN-α and IFN-γ were associated with higher disease activity and inflammatory cytokine production in AS patients. These markers were significantly reduced in responders to TNF inhibitor (TNFi) treatment, suggesting their potential as predictors for treatment response [31]. Conversely, the relative downregulation of anti-inflammatory proteins such as IL-10 further supports the presence of a dysregulated immune response in AS patients. These findings align with previous studies suggesting that AS is marked by an imbalance between pro-inflammatory and anti-inflammatory mediators. In a related study using a high-throughput serum biomarker platform (Olink), 951 unique proteins were assessed in patients with psoriatic arthritis (PsA), psoriasis without arthritis (Pso), AS, and health controls [32]. The study identified 68 differentially expressed proteins in PsA patients compared to healthy controls, with 71% of these differentially expressed proteins also dysregulated in Pso or AS, indicating a shared serum proteomic signature between PsA and Pso. Overall, our results align with these findings, reinforcing the potential of these cytokines as biomarkers for disease monitoring. Compared to previous research, our study offers a more comprehensive proteomic profile, shedding light on additional inflammatory mediators not widely studied in AS.

Notably, we employed a LightGBM classifier integrated with a SHAP analysis to identify the most discriminative proteins between AS and NC groups and to further stratify AS subtypes. This machine learning (ML) approach, combined with traditional statistical methods, strengthens the robustness of our findings and aligns with the growing trend of artificial intelligence applications in biomedical research [33,34,35]. The identified proteins demonstrated high diagnostic accuracy, suggesting their potential utility as disease biomarkers. Future studies should further validate these markers in independent cohorts to assess their reproducibility and clinical applicability.

The positive correlations between the expression levels of inflammatory proteins, such as CXCL9 and NRTN, and BASDAI scores in AS patients underscore their potential as indicators for disease activity. CXCL9 is a significant chemokine in autoimmune arthritis, including AS. Its production is synergistically induced by TNF-α and IFN-γ in human microvascular endothelial cells, and its levels are notably elevated in the synovial fluids of patients with spondyloarthropathies [36]. Previous studies have demonstrated that CXCL9 is elevated in AS patients [37]. Similarly, in this research, we also detected higher serum levels of CXCL9 in AS patients compared to the NC group. Additionally, the NRTN level was also increased in AS and AS_HLA-B2705 patients in this study. To date, there are no studies specifically addressing the role of NRTN in AS. However, this study highlights the potential role of glial-cell-derived neurotrophic factor (GDNF) family ligands (GFLs), including NRTN, in modulating inflammatory responses. Although direct evidence of NRTN’s involvement in AS is limited, studies suggesting its role in repairing epithelial barrier damage after inflammation may provide a basis for exploring its relevance in AS pathogenesis and treatment [38]. Future research should further explore the roles of these candidate proteins and validate their utility in clinical settings for better disease monitoring and tailored therapeutic approaches.

## 5. Conclusions

This study provides a proteomic analysis of the inflammatory protein landscape in HLA-B27-positive AS patients, leveraging advanced proteomic technology to uncover disease-associated molecular signatures. We identified alterations in the expression profiles of inflammatory proteins between AS patients and healthy controls, highlighting the dysregulation of key pro-inflammatory cytokines such as IL-6, IL-17A, TNF, and IFN-γ, and the potential of CXCL9 and NRTN as biomarkers for disease activity. The distribution of HLA-B27 subtypes and the male-to-female ratio observed in our sample align with known prevalence patterns, emphasizing the representativeness of our cohort and the importance of considering gender-specific factors in AS research and treatment. Through the application of the LightGBM classifier, supported by SHAP visualizations, we enhanced the identification of discriminative proteins, demonstrating its value in distinguishing AS patients from healthy controls and in subclassifying AS patients based on HLA-B27 subtypes.

However, this study has several limitations, including the relatively small sample size and cross-sectional design that limit the ability to draw causal inferences and generalize findings. Additionally, our analysis was based on cytokine levels in peripheral blood rather than in the target organs of inflammation, which may not fully reflect the local inflammatory processes occurring in ankylosing spondylitis. Future research should prioritize larger, more diverse cohorts and longitudinal designs, as well as the examination of cytokine expression directly in affected tissues, to validate these findings. Such approaches would enable a more comprehensive understanding of AS pathogenesis, inform the development of targeted treatment strategies, and ultimately improve disease monitoring and patient outcomes.

## Figures and Tables

**Figure 1 biomolecules-15-00516-f001:**
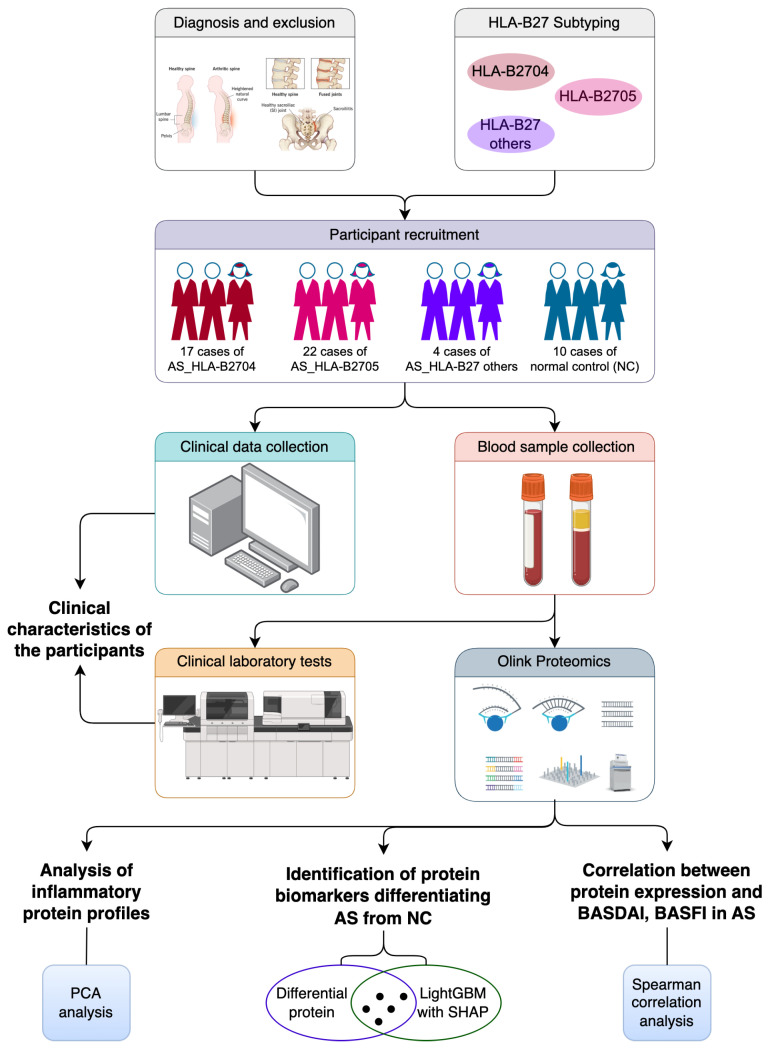
Experimental design and research workflow of this study. HLA-B27: human leukocyte antigen-B*27; AS: ankylosing spondylitis; NC: normal control; PCA: principal component analysis; LightGBM: light gradient boosting machine; SHAP: Shapley Additive Explanations; BASDAI: Bath Ankylosing Spondylitis Disease Activity Index; BASFI: Bath Ankylosing Spondylitis Functional Index.

**Figure 2 biomolecules-15-00516-f002:**
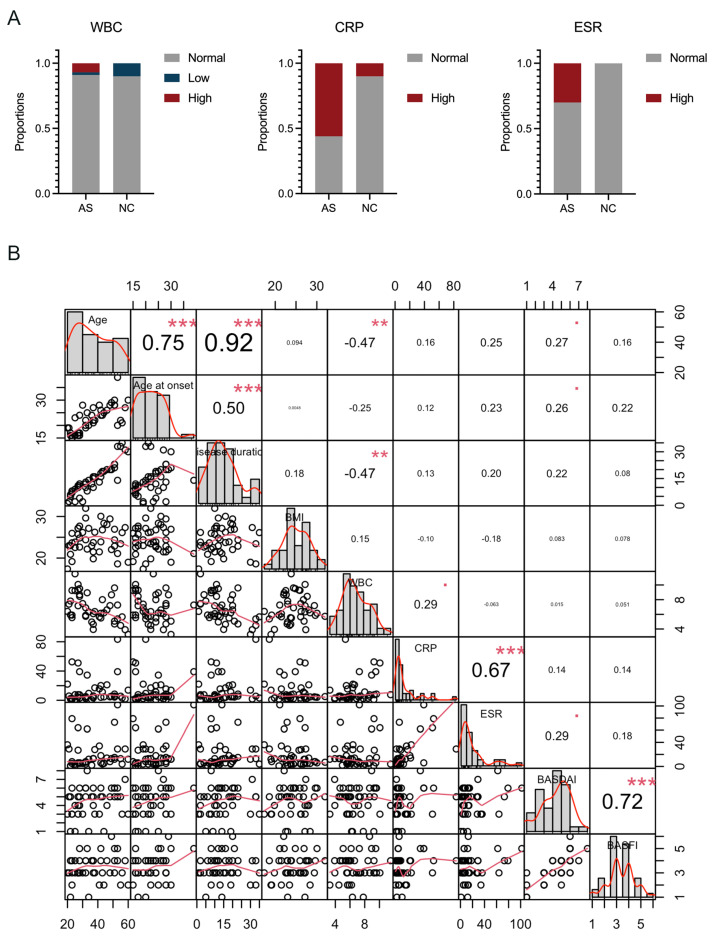
An analysis of the clinical parameters in the participants: (**A**) abnormal rates in WBC, CRP, and ESR tests among AS patients and NC participants; (**B**) a correlation analysis of the clinical parameters among the 43 AS patients was performed using the “chart.Correlation” function in the “PerformanceAnalytics” package in R. WBC: white blood cell; CRP: C-reactive protein; ESR: erythrocyte sedimentation rate. The values within the boxes represent the Spearman correlation coefficients between each pair of parameters, where “**■**” indicates 0.05 < *p* < 0.1, “**” indicates *p* < 0.01, and “***” indicates *p* < 0.001.

**Figure 3 biomolecules-15-00516-f003:**
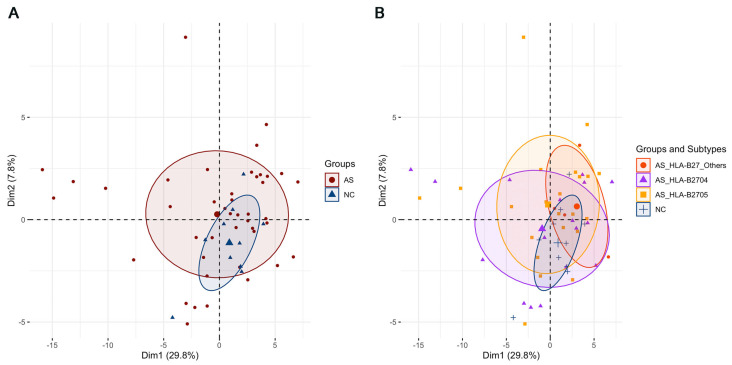
A PCA plot of the samples based on the inflammation protein profiles: (**A**) AS patients as a whole group; (**B**) AS patients with different HLA-B27 subtypes. The PCA plot was generated using the “PCA” function in the “FactoMineR” package in R.

**Figure 4 biomolecules-15-00516-f004:**
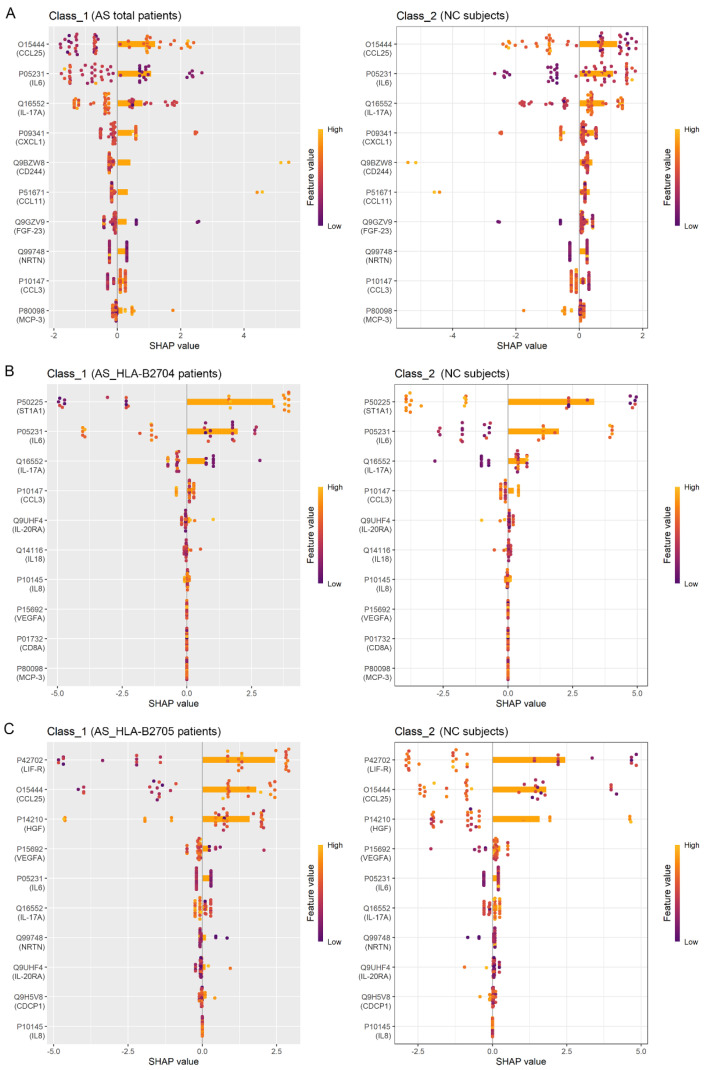
SHAP plots of the top ten proteins identified by the LightGBM classifier for distinguishing different groups: (**A**) AS (Total) vs. NC; (**B**) AS_HLA-B2704 vs. NC; (**C**) AS_HLA-B2705 vs. NC. The LightGBM model was created using the “lightgbm” package in R, and the SHAP plots were generated based on the LightGBM results using the “shapviz” package in R.

**Figure 5 biomolecules-15-00516-f005:**
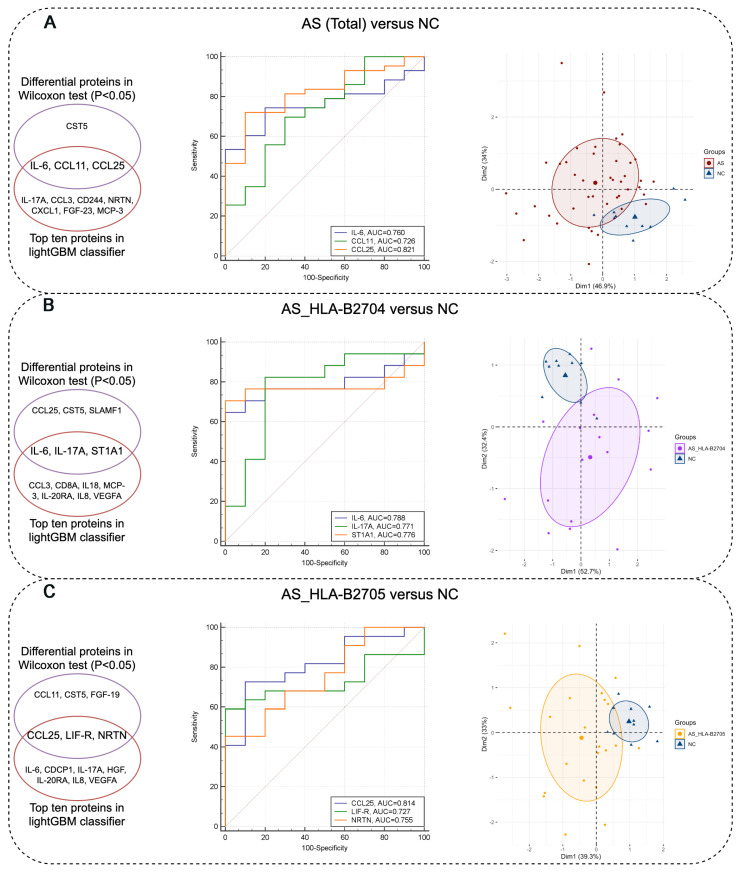
Analysis of selected protein markers for distinguishing between different groups. (**A**) IL-6, CCL11, and CCL25 were identified by both the Wilcoxon test and LightGBM classifier to distinguish AS (Total) from NC. The corresponding ROC analysis and PCA based on these three proteins are shown. (**B**) IL-6, IL-17A, and ST1A1 were identified by both the Wilcoxon test and LightGBM classifier to differentiate AS_HLA-B2704 from NC. The corresponding ROC analysis and PCA based on these three proteins are shown. (**C**) CCL25, LIF-R, and NRTN were identified by both the Wilcoxon test and LightGBM classifier to distinguish AS_HLA-B2705 from NC. The corresponding ROC analysis and PCA based on these three proteins are shown.

**Figure 6 biomolecules-15-00516-f006:**
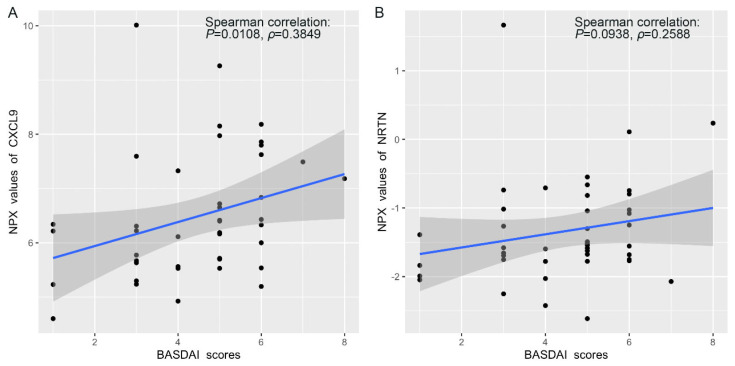
Spearman correlation scatter plots: (**A**) Spearman correlation between the relative expression of CXCL9 and BASDAI scores; (**B**) Spearman correlation between the relative expression of NRTN and BASDAI scores. The plots were generated using “ggplot2” package in R.

**Table 1 biomolecules-15-00516-t001:** Clinical characteristics of participants in the study.

	AS Patients (Number = 43)	NC (Number = 10)
Basic clinical information		
Age, years, (mean ± SD [range])	37.0 ± 11.8 (20~60)	40.1 ± 9.0 (28~53)
Age at onset, years, (mean ± SD [range])	22.5 ± 5.3 (15~39)	–––
Disease duration, years, (mean ± SD, range)	14.5 ± 8.8 (1~35)	–––
Gender (number of Male/Female)	30/13	7/3
BMI, kg/m^2^, (mean ± SD [range])	24.4 ± 3.3 (17.4~32.0)	23.5 ± 4.4 (18.0~33.9)
Drug intervention (number of Yes/No)	32/11	–––
Clinical laboratory tests		
HLA-B27+ (number, %)	43, 100%	–––
HLA-B27 Subtypes		–––
HLA-B2704 (number, %)	17, 39.5%	–––
HLA-B2705 (number, %)	22, 51.2%	–––
Others (number, %)	4, 9.3%	–––
WBC, 10^9, (mean ± SD [range])	6.8 ± 1.9 (3.26~11.52)	5.9 ± 1.8 (3.03~8.43)
CRP, mg/L, (mean ± SD [range])	12.8 ± 17.2 (0.50~83.32) **	2.3 ± 2.0 (0.20~5.75)
ESR, mm/H, (mean ± SD [range])	19.4 ± 23.3 (1~102)	8.4 ± 4.1 (1~14)
Index scores		
BASDAI (mean ± SD [range])	4.4 ± 1.7 (1~8)	–––
BASFI (mean ± SD [range])	3.4 ± 1.1 (1~6)	–––

Note: Reference range: WBC (3.5–9.5) ´109/L; CRP (0–5) mg/L; ESR male (0–15) mm/H, ESR female (0–20) mm/H; “**” represents *p* < 0.01 in the comparison of AS and NC subjects in Wilcoxon rank sum test.

**Table 2 biomolecules-15-00516-t002:** A differential protein analysis using the Wilcoxon rank-sum test.

	Assay	OlinkID	UniProt	Disease (Mean)	NC (Mean)	*p*_Value
AS (Total) vs. NC	CCL25	OID00551	O15444	5.52	6.17	0.0011
	IL-6	OID00482	P05231	2.80	1.58	0.0097
	CST5	OID00491	P28325	5.88	6.29	0.0266
	CCL11	OID00505	P51671	6.55	6.93	0.0266
AS_HLA-B2704 vs. NC	CCL25	OID00551	O15444	5.44	6.17	0.0022
	IL-6	OID00482	P05231	2.72	1.58	0.0129
	CST5	OID00491	P28325	5.85	6.29	0.0459
	ST1A1	OID00557	P50225	7.42	8.91	0.0175
	SLAMF1	OID00502	Q13291	1.00	0.63	0.0459
	IL-17A	OID00485	Q16552	0.98	0.43	0.0203
AS_HLA-B2705 vs. NC	CCL25	OID00551	O15444	5.56	6.17	0.0040
	FGF-19	OID00545	O95750	5.39	6.18	0.0385
	CST5	OID00491	P28325	5.86	6.29	0.0385
	LIF-R	OID00511	P42702	2.78	3.00	0.0428
	CCL11	OID00505	P51671	6.53	6.93	0.0249
	NRTN	OID00548	Q99748	−1.31	−1.80	0.0222

Note: The mean expression values in the table are based on the normalized protein expression (NPX) values.

## Data Availability

The original contributions presented in this study are included in the article/Appendix A. Further inquiries can be directed to the corresponding authors.

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
