# Peer review of "Proteomic Profiling of Inflammatory Protein Dysregulation in HLA-B27-Positive Ankylosing Spondylitis: Molecular Signatures and Potential Biomarkers"

_biomolecules, 2025, doi:10.3390/biom15040516_

Round 1

Reviewer 1 Report

Comments and Suggestions for Authors

This is an interesting paper on the profile of inflammatory proteins in HLA-B27 positive AS patients, contributing to the current knowledge of AS-associated molecular signatures. This study shows a dysregulation of key pro-inflammatory cytokines such as IL6, IL-17A, TNF and IFN-γ in AS patients compared to normal controls and the potential of CXCL9 and NRTN as biomarkers of disease activity. This is a very interesting and original article in the search for new biomarkers to improve both the classification and treatment of AS patients.

The results are clearly presented and organized according to an adequate argumentative strategy. However, there are a few issues relative to the Introduction section and the Figures that if addressed by the authors will further enhance the clarity of the manuscript.

-  The Introduction includes irrelevant information regarding the main line of the paper. This section includes too much information that should be suppressed or summarised to make the text more fluid. Since this is not a review article, only the basic information should be presented in order to provide a context for the results obtained. Examples of content that would be subject to removal include :

Lines 44-45. “ Meta-analyses support these findings, showing the lowest prevalence in Africa and higher prevalence in Northern Arctic communities, North America, and Europe”

Lines 49-51. “ This early sex discrepancy is attributed to more severe radiographic findings in males, which were more easily detectable. However, with the advent of advanced imaging techniques, diagnostic rates are becoming more equal between sexes”

Lines 57-59: “ The molecular basis of this association continues to attract  significant interest, as unraveling the precise mechanisms may reveal key pathways involved in immune dysregulation and chronic inflammation.”

Lines 60-62: “As of 2013, HLA-B27 exhibits significant genetic polymorphism, with 105 known subtypes (HLA- 61 B27:01 to HLA-B27:106) encoded by 132 alleles.”

Lines 70-72. “ Investigating these disparities provides an opportunity to explore how genetic variations influence immune-mediated processes,thereby contributing to our understanding of the biomolecular basis of AS.”

Lines 75-77: “ Achieving these goals necessitates early detection and intervention, guiding patients on proper posture, and encouraging appropriate exercise. Thus, analyzing the inflammation conditions of AS is crucial due to its impact on the disease's progression”

- I consider this last sentence of the Introduction to be overambitious, as the results do not allow us to infer the mechanism by which HLA27 is involved in the disease. (lines 110. “Moreover, this study's findings may contribute to a more nuanced understanding of the role of HLA-B27 in the broader spectrum of spondyloarthropathies.”)

-Figure 1: Please include in the figure caption the description of the abbreviations shown in the figure.

-Figure 2: Please add the description of the abbreviations WBC, CRP, ESR in the caption. In addition, the size of some of the numbers in the figure is so small that they cannot be read. It would be advisable to unify the size of the numbers in the figure.

-Lines 183-185 “The enrichment of upregulated proteins…” The same text repeated in discussion (320-322). Sentences should not be repeated literally in the text, it is redundant.

- Figure 5: please increase the size of the graphics included in the figure. They are illegible

-Lines 251-253. This sentence is not appropriate for Results, but for discussion, as it is an interpretation of the results that does not correspond to the section

- Lines: 288-291 It looks like the information in these lines lacks a bibliographic reference.

- Line 296: I am not entirely sure what they mean by “in IL-17-enriched peripheral blood mononuclear cells”.

- Lines 340-344 I misunderstand the contribution of the information included by the authors on NRTN in these lines. I think it confuses the reader and does not add anything to the AS.

Author Response

Reviewer 1

This is an interesting paper on the profile of inflammatory proteins in HLA-B27 positive AS patients, contributing to the current knowledge of AS-associated molecular signatures. This study shows a dysregulation of key pro-inflammatory cytokines such as IL6, IL-17A, TNF and IFN-γ in AS patients compared to normal controls and the potential of CXCL9 and NRTN as biomarkers of disease activity. This is a very interesting and original article in the search for new biomarkers to improve both the classification and treatment of AS patients.

The results are clearly presented and organized according to an adequate argumentative strategy. However, there are a few issues relative to the Introduction section and the Figures that if addressed by the authors will further enhance the clarity of the manuscript.

Response: We sincerely thank all four reviewers for their insightful comments and valuable suggestions, which have significantly contributed to enhancing the quality of our manuscript. We have carefully addressed each comment point by point and made corresponding revisions throughout the manuscript. In particular, we have streamlined the content by removing certain sections from the Introduction, Results, and Discussion to improve clarity and focus on the key findings. Additionally, we have included more detailed descriptions of the procedures in the Methods section. In the Conclusion part, the corresponding contents are added and more limitations of this research are put forward. Specifically, Figure 1 (illustrating the experimental design and research workflow), Figure 3 (PCA plot based on the 81 inflammation-related protein profiles), Figure 5 (analysis of selected protein markers distinguishing between different groups), Table 2, and their associated descriptions have been re-analyzed or re-generated in the revised manuscript. Furthermore, the KEGG analysis was removed to resolve any potential confusion it may have caused. These adjustments, we believe, enhance the readability, coherence, and impact of our findings. All newly added or modified content has been highlighted in red font. We hope that the revised version meets the expectations of the reviewers and the standards of the journal.

-  The Introduction includes irrelevant information regarding the main line of the paper. This section includes too much information that should be suppressed or summarized to make the text more fluid. Since this is not a review article, only the basic information should be presented in order to provide a context for the results obtained. Examples of content that would be subject to removal include:

Lines 44-45. “Meta-analyses support these findings, showing the lowest prevalence in Africa and higher prevalence in Northern Arctic communities, North America, and Europe”

Lines 49-51. “This early sex discrepancy is attributed to more severe radiographic findings in males, which were more easily detectable. However, with the advent of advanced imaging techniques, diagnostic rates are becoming more equal between sexes”

Lines 57-59: “The molecular basis of this association continues to attract significant interest, as unraveling the precise mechanisms may reveal key pathways involved in immune dysregulation and chronic inflammation.”

Lines 60-62: “As of 2013, HLA-B27 exhibits significant genetic polymorphism, with 105 known subtypes (HLA- 61 B27:01 to HLA-B27:106) encoded by 132 alleles.”

Lines 70-72. “Investigating these disparities provides an opportunity to explore how genetic variations influence immune-mediated processes, thereby contributing to our understanding of the biomolecular basis of AS.”

Lines 75-77: “Achieving these goals necessitates early detection and intervention, guiding patients on proper posture, and encouraging appropriate exercise. Thus, analyzing the inflammation conditions of AS is crucial due to its impact on the disease's progression”

- I consider this last sentence of the Introduction to be overambitious, as the results do not allow us to infer the mechanism by which HLA27 is involved in the disease. (line 110. “Moreover, this study's findings may contribute to a more nuanced understanding of the role of HLA-B27 in the broader spectrum of spondyloarthropathies.”)

Answer: We thank the reviewer very much for the careful review, we have deleted the irrelevant information and modified several sentences in the Introduction.

-Figure 1: Please include in the figure caption the description of the abbreviations shown in the figure.

Answer: We thank the reviewer very much and added the description of the abbreviations in Figure 1.

-Figure 2: Please add the description of the abbreviations WBC, CRP, ESR in the caption. In addition, the size of some of the numbers in the figure is so small that they cannot be read. It would be advisable to unify the size of the numbers in the figure.

Answer: We thank the reviewer very much and added the description of the abbreviations WBC, CRP, ESR in the caption. As for Figure 2B, this dot plot was performed and generated using the “chart.Correlation” function in the “PerformanceAnalytics” package in R. We would like to clarify that the current font size is the default and standard setting, which reflects the magnitude of the correlation coefficients—larger fonts indicate stronger correlations, while smaller fonts suggest weaker or statistically non-significant correlations. Therefore, adjusting the font size uniformly would compromise the interpretative value of the figure. However, we ensure that the figure is of high resolution, allowing readers to clearly view the details by zooming in.

-Lines 183-185 “The enrichment of upregulated proteins…” The same text repeated in discussion (320-322). Sentences should not be repeated literally in the text, it is redundant.

Answer: We thank the reviewer for the careful review. Based on a comprehensive comments and suggestions provided by all four reviewers, we have decided to remove the section on the KEGG pathway enrichment analysis of the proteins, including the corresponding methods, results, and discussion parts. This adjustment was made to enhance the clarity and coherence of the manuscript, making it more streamlined and focused.

- Figure 5: please increase the size of the graphics included in the figure. They are illegible.

Answer: We thank the reviewer for the comments. This figure (Figure 4 in the revised version) is presented as a composite to showcase the overall results. It is a high-resolution image, and could zoom in for a clearer view of the details. We have ensured that the image quality is sufficient for close inspection without loss of clarity.

-Lines 251-253. This sentence is not appropriate for Results, but for discussion, as it is an interpretation of the results that does not correspond to the section.

Answer: We thank the reviewer very much for the careful review. The original content “These correlations highlight the potential of these proteins as biomarkers for disease activity and could aid in developing personalized treatment strategies” was deleted in the Results section and integrated into the Discussion section.

- Lines: 288-291 It looks like the information in these lines lacks a bibliographic reference.

Answer: We thank the review for the careful review and added references to the content. Pro-inflammatory cytokines, including IL-6, IL-17A, TNF, and IFN-γ, are relatively activated in AS, underscoring their roles in the disease's inflammatory cascade[27, 28]. IL-6 is known for promoting inflammation and autoimmunity, while IL-17A is critical in the pathogenesis of several autoimmune diseases, including AS[29].

- Line 296: I am not entirely sure what they mean by “in IL-17-enriched peripheral blood mononuclear cells”.

Answer: We thank the reviewer for the queries. The “IL-17-enriched peripheral blood mononuclear cells” is the way it was written in the reference #16. The "IL-17-enriched peripheral blood mononuclear cells (PBMCs)" refer to PBMCs that were stimulated to enhance the proportion of IL-17-producing cells, particularly Th17 cells, which are known to play a crucial role in AS pathogenesis.

- Lines 340-344 I misunderstand the contribution of the information included by the authors on NRTN in these lines. I think it confuses the reader and does not add anything to the AS.

Answer: We thank the reviewer for the valuable comments. We acknowledge that there is limited research directly linking NRTN to AS. Our intention in including this information was to discuss relevant studies that highlight the role of NRTN in inflammatory diseases more broadly, which we believe could provide some useful context for exploring potential therapeutic targets in AS. Besides, we have revised the text to improve clarity and to help readers better understand the relevance of NRTN in the context of our study.

Reviewer 2 Report

Comments and Suggestions for Authors

This research paper reports a proteomic analysis of clinical material from HLA-B27 Positive Ankylosing Spondylitis. The authors discuss correlation between AS pathogenesis and genetic predisposition (HLA-B27). Experimental parts describe collection and analysis of 53 peripheral blood samples (43 AS and 10 healthy controls) and their testing in a clinical laboratory setting for WBC count, and also for CRP levels, and ESR. Additional tests performed by the authors were HLA-B27 subtyping by PCR, and the use of proximity extension assay to assess and quantify the levels of 92 inflammation related protein markers. The reported research is adequate and the manuscript reads well, but there are issues with data analysis and to a lesser degree with the presentation.

Comments:

The Introduction appears relevant but excessive length-wise, it should be shortened and streamlined. Most of the text in this section does not add much value to the manuscript and may be removed. Instead, the section should focus on (1) justification of the decision to analyse inflammation markers and (2) justification of the choice of Olink Target 96 inflammation panel, rather than any other proteomic method, from a plethora of proteomics methodologies available.

The Results section describes substantial research effort, but the presentation is in need of improvement. Figure 2 is too busy and not well formatted. I assume it was pasted directly from the Performance Analytics package, which could make reformatting difficult. There is no need to show the multitude of insignificant relationships on this busy and not very helpful figure. It would be best to remove this figure from the manuscript and display it in the Supplementary Materials only. Instead, a few properly formatted cases of significant positive correlations should be displayed using clear, simple graphs, which can easily be generated in Excel or any other basic data analysis package. Or even a simple small Table with numerical values only to summarise these positive and significant cases would be sufficient. Furthermore, the Results section should interpret the data shown. Just showing the graphs is not sufficient. What have the authors concluded from those correlations? How do these helped to progress research to the next stage? Are these correlations really needed? What have the authors learned from these?

Contrary ot the authors statement (lines 161-2), I can; not see any separation in Fig.3. and contrary to statement in lines 164-5, I do not see any obvious clustering in Fig.3. I can't see any subtype-specific protein expression patterns in this figure.

The statement in lines 175-177 about ignoring statistical significance is rather irresponsible and indicates a disregard for the most basic principles of statistics. Authors misinterpret their data which invalidate that section (2.2.), the conclusions and the subsequent steps. The data should be re-analysed and the section rewritten.

Section 2.3 (starting line 194). has issues also. It is not acceptable generally to consider P<0.1 as 'significant. Authors should consider P<0.05 or better P<0.01. There are a few proteins with very low P and I see no need to cut corners here and artificially increase the list. More does not mean better. The data should be re-analysed and the text rewritten.

Arbitrary merging of the Wilcoxon and the LightGBM classifier makes little sense (line 214). Instead, these two analyses should have been compared and contracted to each another. Since the results of the two analyses are rather different, it makes the reported results doubtful.  

The ROC curves are not particularly impressive. Please refer to my comments about statistical assessment (above). Authors should discard and not consider proteins which are not significantly different and repeat the analysis. This might potentially improve the ROC analysis. Principal components in Fig. 6 are not revealing any distinct clusters and are not interpreted in the text. The data should be re-analysed and the section rewritten.

Section 2.3 (starting line 243). incorrect section number. The two proteins reported to have a small positive correlation with the BASDAI and BASFI scores do not seem to feature appear in the previous two analyses, which raises questions about the internal consistency of the reported results.

The Discussion section is too lengthy and unfocused. The authors repeat much of their results instead of critically evaluating them. The authors are encouraged to move text reporting results to the Results section, reduce the length of the Discussion, and ensure that their findings are carefully evaluated. This should include, for example, comparing the outcomes to relevant studies in the literature.

Materials and Methods section 4.7. Procedures used to QC and to normalise the recorded Ct data should be reported.

Author Response

Reviewer 2

This research paper reports a proteomic analysis of clinical material from HLA-B27 Positive Ankylosing Spondylitis. The authors discuss correlation between AS pathogenesis and genetic predisposition (HLA-B27). Experimental parts describe collection and analysis of 53 peripheral blood samples (43 AS and 10 healthy controls) and their testing in a clinical laboratory setting for WBC count, and also for CRP levels, and ESR. Additional tests performed by the authors were HLA-B27 subtyping by PCR, and the use of proximity extension assay to assess and quantify the levels of 92 inflammation related protein markers. The reported research is adequate and the manuscript reads well, but there are issues with data analysis and to a lesser degree with the presentation.

Response: We sincerely thank all four reviewers for their insightful comments and valuable suggestions, which have significantly contributed to enhancing the quality of our manuscript. We have carefully addressed each comment point by point and made corresponding revisions throughout the manuscript. In particular, we have streamlined the content by removing certain sections from the Introduction, Results, and Discussion to improve clarity and focus on the key findings. Additionally, we have included more detailed descriptions of the procedures in the Methods section. In the Conclusion part, the corresponding contents are added and more limitations of this research are put forward. Specifically, Figure 1 (illustrating the experimental design and research workflow), Figure 3 (PCA plot based on the 81 inflammation-related protein profiles), Figure 5 (analysis of selected protein markers distinguishing between different groups), Table 2, and their associated descriptions have been re-analyzed or re-generated in the revised manuscript. Furthermore, the KEGG analysis was removed to resolve any potential confusion it may have caused. These adjustments, we believe, enhance the readability, coherence, and impact of our findings. All newly added or modified content has been highlighted in red font. We hope that the revised version meets the expectations of the reviewers and the standards of the journal.

The Introduction appears relevant but excessive length-wise, it should be shortened and streamlined. Most of the text in this section does not add much value to the manuscript and may be removed. Instead, the section should focus on (1) justification of the decision to analyze inflammation markers and (2) justification of the choice of Olink Target 96 inflammation panel, rather than any other proteomic method, from a plethora of proteomics methodologies available.

Answer: We thank the reviewer very much for the valuable comments. In the revised version, we have deleted the irrelevant information and modified several sentences in the Introduction.

The Results section describes substantial research effort, but the presentation is in need of improvement. Figure 2 is too busy and not well formatted. I assume it was pasted directly from the Performance Analytics package, which could make reformatting difficult. There is no need to show the multitude of insignificant relationships on this busy and not very helpful figure. It would be best to remove this figure from the manuscript and display it in the Supplementary Materials only. Instead, a few properly formatted cases of significant positive correlations should be displayed using clear, simple graphs, which can easily be generated in Excel or any other basic data analysis package. Or even a simple small Table with numerical values only to summarize these positive and significant cases would be sufficient. Furthermore, the Results section should interpret the data shown. Just showing the graphs is not sufficient. What have the authors concluded from those correlations? How do these helped to progress research to the next stage? Are these correlations really needed? What have the authors learned from these?

Answer: We thank the reviewer very much for the careful and insightful comments. Indeed, as the reviewer mentioned, the Figure 2B dot plot was performed and generated using the “chart.Correlation” function in the “PerformanceAnalytics” package in R (with the default and standard settings). This plot reflects the magnitude of the correlation coefficients—larger fonts indicate stronger correlations, while smaller fonts suggest weaker or statistically non-significant correlations. After thorough consideration, we have decided to retain Figure 2B in the main text. Our intention in presenting Figure 2B was to provide a comprehensive overview of the correlations among clinical information, which may contribute to a better understanding of the interconnections within the clinical data.

Contrary on the authors statement (lines 161-2), I can; not see any separation in Fig.3. and contrary to statement in lines 164-5, I do not see any obvious clustering in Fig.3. I can't see any subtype-specific protein expression patterns in this figure.

Answer: We thank the reviewer for the careful review. We acknowledge that the separation and clustering patterns in Figure 3 were not very pronounced. Our primary goal with this analysis was to gain an initial overview of the entire protein profiling distribution across different groups, which could help us identify potential trends worth exploring further. This preliminary PCA analysis served as a starting point for narrowing down potential biomarkers. In the revised manuscript, we have clarified this and rephrased the context accordingly, as the reviewer observed.

The statement in lines 175-177 about ignoring statistical significance is rather irresponsible and indicates a disregard for the most basic principles of statistics. Authors misinterpret their data which invalidate that section (2.2.), the conclusions and the subsequent steps. The data should be re-analyzed and the section rewritten.

Answer: We thank the reviewer for the careful review and valuable suggestion. Based on a comprehensive comments and suggestions provided by all four reviewers, we have decided to remove the section on the KEGG pathway enrichment analysis of the proteins, including the corresponding methods, results, and discussion parts. This adjustment was made to enhance the clarity and coherence of the manuscript, making it more streamlined and focused.

Section 2.3 (starting line 194). has issues also. It is not acceptable generally to consider P<0.1 as 'significant. Authors should consider P<0.05 or better P<0.01. There are a few proteins with very low P and I see no need to cut corners here and artificially increase the list. More does not mean better. The data should be re-analyzed and the text rewritten.

Answer: We sincerely thank the reviewer for the valuable suggestion. Indeed, as the reviewer pointed out, in the original manuscript, we used P < 0.1 as the threshold for significance primarily to obtain a more protein list. In the revised version, we have taken the reviewer's suggestion into account and have re-analyzed the data using P < 0.05 to identify proteins with statistically significant differences. The context and Table 2 were revised accordingly.

Arbitrary merging of the Wilcoxon and the LightGBM classifier makes little sense (line 214). Instead, these two analyses should have been compared and contracted to each another. Since the results of the two analyses are rather different, it makes the reported results doubtful.  

Answer: We sincerely appreciate the reviewer's insightful comments. In this study, in addition to traditional statistical methods such as the Wilcoxon test, we introduced machine learning-based approaches like the LightGBM classifier. This reflects the current trend of integrating AI technologies into biomedical data analysis to enhance the robustness and reliability of research findings. The use of both methods aimed to provide a comprehensive analysis and to cross-validate the results. Moreover, the overlapping results obtained from these two approaches demonstrated promising potential for disease classification and diagnosis, as illustrated in Figure 5. This cross-validation supports the credibility of our findings and highlights the practical applicability of combining traditional and machine learning methods in biomarker discovery.

The ROC curves are not particularly impressive. Please refer to my comments about statistical assessment (above). Authors should discard and not consider proteins which are not significantly different and repeat the analysis. This might potentially improve the ROC analysis. Principal components in Fig. 6 are not revealing any distinct clusters and are not interpreted in the text. The data should be re-analyzed and the section rewritten.

Answer: We sincerely appreciate the reviewer's valuable suggestions. Following your recommendation, we have reanalyzed the data and revised the manuscript accordingly. As suggested, we focused on significantly different proteins (P<0.05 in Wilcoxon test), which indeed enhanced the performance of the ROC analysis and improved the PCA results. For example, in the updated Figure 5, the new ROC curves demonstrate a better diagnostic capability (AUC>0.7 in all curves). Moreover, compared to the initial PCA plot (Figure 2), the revised PCA analysis using the filtered protein biomarkers reveals a much clearer separation between each two comparison groups, highlighting the improved discriminative power of the selected markers.

Section 2.3 (starting line 243). incorrect section number. The two proteins reported to have a small positive correlation with the BASDAI and BASFI scores do not seem to feature appear in the previous two analyses, which raises questions about the internal consistency of the reported results.

Answer: We thank the reviewer for the comment. The two proteins showing some positive correlation with the BASDAI and BASFI scores did not appear in the previous two analyses because these analyses focused on identifying proteins with significant differences between the comparison groups rather than correlating proteins directly with clinical indices. The absence of these proteins in the previous analyses does not imply a lack of internal consistency but rather reflects the different analytical approaches and objectives. In our study, we aimed to provide a comprehensive analysis by exploring both differential expression and potential clinical correlations.

The Discussion section is too lengthy and unfocused. The authors repeat much of their results instead of critically evaluating them. The authors are encouraged to move text reporting results to the Results section, reduce the length of the Discussion, and ensure that their findings are carefully evaluated. This should include, for example, comparing the outcomes to relevant studies in the literature.

Answer: We thank the reviewer very much for the valuable comments, and have updated the Discussion accordingly.

Materials and Methods section 4.7. Procedures used to QC and to normalize the recorded Ct data should be reported.

Answer: We thank the reviewer for the careful review, we have added the detailed information about procedures used to QC and to normalize the recorded Ct data in the revised version.

Reviewer 3 Report

Comments and Suggestions for Authors

Thank you for the possibility to review the paper: „ Proteomic Profiling of Inflammatory Protein Dysregulation in HLA-B27 Positive Ankylosing Spondylitis: Molecular Signatures and Potential Biomarkers“.

It deals in very important subject however I have some major issues with it.

The inclusion group has very high CRP and SE, and I believe that this is an inclusion bias.

AS is not known for such an increase in CRP, perhaps some patients with lower CRP were excluded.

Also, exclusion criteria were very wide: DMARD usage, biologic therapy, high dose corticosteroids, among others.

This leaves us with newly discovered AS patients.

The authors suggested that this is a representative population of AS in China. They reported in the Introduction that HLA–B2704 is the most prevalent in the Chinese population. In this paper, 51% were HLA B2705, and only 39% of HLA–B2704 , so it is not completely representative.

This means that the results part of the paper could be influenced by not including all of the patients but just the once with higher disease activity.

The results section is written in language that is not easy to read, especially the part KEGG pathway enrichment analysis. I think they should omit this part from the paper. The data from Figure 4 shows the same proteins both downregulated and upregulated. The results are written in a way that favours author's hypothesis without providing substantial evidence.

The main results are also very controversial, TNFα and IFNγ are non-significantly elevated and IL 17 is borderline significant (0.0568) in the total HLA B27 group, the only difference is in IL 6 value which can be explained by choosing the patients with elevated CRP. IL 6 is known to be elevated in patients with increased acute phase proteins like CRP.

Furthermore, results from R package like Facto Miner, and Shapviz, are unreadable to me and can be explained in a multitude of ways. Figure 2 is also confusing for me.

I believe the authors should simplify the ways of presenting the main results, and be academically true and say, the evidence are not confirming our results but the trends are positive toward future proof of our hypothesis.

It can be that my expertise in this area is very bad and You can disregard my review but I believe that no one can repeat this experiment, because it is not clear which patients were chosen, if there were some bias, and whether the data that IL 6 elevation is pnly the results of increased CRP values.

Maybe upon major revision it will be easier to follow the results, without using tests like KEGG which are very confusing and do not add to value of this paper but add the mystical component to the very hard work done by the scientific team.

I believe there is a value of this paper if all figures were simplified or made into tables, and discussion was written only in the hard evidence obtained by the statistically significant results not borderline results.

Cordial regards.

Author Response

Reviewer 3

Thank you for the possibility to review the paper: “Proteomic Profiling of Inflammatory Protein Dysregulation in HLA-B27 Positive Ankylosing Spondylitis: Molecular Signatures and Potential Biomarkers”.

It deals in very important subject however I have some major issues with it.

Response: We sincerely thank all four reviewers for their insightful comments and valuable suggestions, which have significantly contributed to enhancing the quality of our manuscript. We have carefully addressed each comment point by point and made corresponding revisions throughout the manuscript. In particular, we have streamlined the content by removing certain sections from the Introduction, Results, and Discussion to improve clarity and focus on the key findings. Additionally, we have included more detailed descriptions of the procedures in the Methods section. In the Conclusion part, the corresponding contents are added and more limitations of this research are put forward. Specifically, Figure 1 (illustrating the experimental design and research workflow), Figure 3 (PCA plot based on the 81 inflammation-related protein profiles), Figure 5 (analysis of selected protein markers distinguishing between different groups), Table 2, and their associated descriptions have been re-analyzed or re-generated in the revised manuscript. Furthermore, the KEGG analysis was removed to resolve any potential confusion it may have caused. These adjustments, we believe, enhance the readability, coherence, and impact of our findings. All newly added or modified content has been highlighted in red font. We hope that the revised version meets the expectations of the reviewers and the standards of the journal.

The inclusion group has very high CRP and ESR, and I believe that this is an inclusion bias.

AS is not known for such an increase in CRP, perhaps some patients with lower CRP were excluded.

Also, exclusion criteria were very wide: DMARD usage, biologic therapy, high dose corticosteroids, among others.

This leaves us with newly discovered AS patients.

Answer: We thank the reviewer for the valuable comments and suggestions. We would like to clarify that our inclusion criteria were not specifically focused on selecting patients with active inflammation. The relatively high CRP and ESR levels observed in the inclusion group reflect the characteristics of the enrolled AS patients rather than an intentional selection bias. As for the exclusion criteria, such as the use of DMARDs, biologic therapy, and high-dose corticosteroids, these were applied to minimize potential confounding effects from medications that could influence protein expression profiles, thereby ensuring that the observed results more accurately represent the disease's underlying protein signatures. Consequently, the study population mainly consisted of untreated AS patients, which we believe enhances the reliability of the identified biomarkers.

The authors suggested that this is a representative population of AS in China. They reported in the Introduction that HLA–B2704 is the most prevalent in the Chinese population. In this paper, 51% were HLA B2705, and only 39% of HLA–B2704, so it is not completely representative.

Answer: We thank the reviewer very much for the careful review. In the revised version, we deleted this section, to make the Discussion section more concise and focus on the key points

This means that the results part of the paper could be influenced by not including all of the patients but just the once with higher disease activity.

The results section is written in language that is not easy to read, especially the part KEGG pathway enrichment analysis. I think they should omit this part from the paper. The data from Figure 4 shows the same proteins both downregulated and upregulated. The results are written in a way that favours author's hypothesis without providing substantial evidence.

Answer: We thank the reviewer very much for the valuable comments. Based on a comprehensive comments and suggestions provided by all four reviewers, we have decided to remove the section on the KEGG pathway enrichment analysis of the proteins, including the corresponding methods, results, and discussion parts. This adjustment was made to enhance the clarity and coherence of the manuscript, making it more streamlined and focused.

The main results are also very controversial, TNFα and IFN-γ are non-significantly elevated and IL 17 is borderline significant (0.0568) in the total HLA B27 group, the only difference is in IL 6 value which can be explained by choosing the patients with elevated CRP. IL 6 is known to be elevated in patients with increased acute phase proteins like CRP.

Answer: We thank the reviewer for the careful review. As the reviewer pointed out, the elevation of IL-6 may indeed be influenced by the selection of patients with elevated CRP levels, given IL-6's known association with acute-phase responses. However, our primary aim was not to focus solely on individual cytokines but rather to explore a broader profile of inflammation-related proteins using both traditional statistical methods and machine learning approaches to identify potential biomarkers for AS. This integration of machine learning approaches with traditional statistical methods not only enhances the robustness of our findings but also aligns with the emerging trend of applying artificial intelligence (AI) in biomedical research. We believe that this comprehensive approach provides valuable insights into the inflammatory landscape of AS, even if some individual cytokines did not reach statistical significance.

Furthermore, results from R package like Facto Miner, and Shapviz, are unreadable to me and can be explained in a multitude of ways. Figure 2 is also confusing for me.

Answer: We appreciate the reviewer’s comments. Our intention in including detailed information about the statistical and analytical tools, such as the FactoMineR and Shapviz packages in R, was to enhance the transparency and reproducibility of our data analysis. By explicitly stating the tools and methods used, we aimed to provide readers with a clear understanding of the analytical pipeline, thereby increasing the credibility and reliability of the results. As for Figure 2B, this dot plot was performed and generated using the “chart.Correlation” function in the “PerformanceAnalytics” package in R. We would like to clarify that the current font size is the default and standard setting, which reflects the magnitude of the correlation coefficients—larger fonts indicate stronger correlations, while smaller fonts suggest weaker or statistically non-significant correlations.

I believe the authors should simplify the ways of presenting the main results, and be academically true and say, the evidence is not confirming our results but the trends are positive toward future proof of our hypothesis.

Answer: We thank the reviewer very much for the suggestion. As the reviewer mentioned, we have simplified or deleted some content in the Results section, to make the results better displayed.

It can be that my expertise in this area is very bad and You can disregard my review but I believe that no one can repeat this experiment, because it is not clear which patients were chosen, if there were some bias, and whether the data that IL 6 elevation is only the results of increased CRP values.

Answer: Overall, we sincerely thank the reviewer for their valuable feedback and constructive suggestions, which have significantly helped us improve the quality of our manuscript. We understand the concerns regarding the reproducibility of our findings and the potential biases in patient selection. To address this, we would like to clarify that the raw data, along with the analysis scripts used at each step of the study, can be made available upon request by contacting the corresponding author. This is intended to ensure that our results can be accurately reproduced and verified by other researchers. Regarding the potential biases in patient selection, we acknowledge this as an important point and will carefully consider these aspects in future studies. We greatly appreciate the reviewer's insights and hope that these clarifications help address the concerns raised.

Maybe upon major revision it will be easier to follow the results, without using tests like KEGG which are very confusing and do not add to value of this paper but add the mystical component to the very hard work done by the scientific team.

Answer: We thank the reviewer for the careful review. Based on a comprehensive comments and suggestions provided by all four reviewers, we have decided to remove the section on the KEGG pathway enrichment analysis of the proteins, including the corresponding methods, results, and discussion parts. This adjustment was made to enhance the clarity and coherence of the manuscript, making it more streamlined and focused.

I believe there is a value of this paper if all figures were simplified or made into tables, and discussion was written only in the hard evidence obtained by the statistically significant results not borderline results.

Answer: We sincerely appreciate the reviewer's valuable suggestions. In response, we have made substantial revisions to the manuscript, particularly in the Results section. We have streamlined the results by retaining only those findings that are statistically significant, to enhance the clarity and impact of our conclusions. Additionally, some figures and tables have been re-analyzed and reformatted to improve readability and convey the information more effectively. Moreover, the Introduction and Discussion sections have been accordingly modified and refined to focus more clearly on the key findings. Thank you again for your insightful feedback.

Reviewer 4 Report

Comments and Suggestions for Authors

Major points

  1. Internationally the term Ankylosing Spondylitis has been replaces by radiographic axal spondylarthritis and I think that authors should make changes throughout the both the title and manuscript. For those readers, who may not be aware of this fact a sentence early in the introduction may be appropriate to clarify this and put the terminology into a perspective.
  2. Statistical methods should be checked. Specifically I am worried about the low p-values in Table 2 considering the very low number of samples.
  3. Overall I think that the authors address an interesting subject, but stretches the analyses to far (considering the low number of patients and in particular controls). For example analyzing the relation to clinical parameters like BASDAI and BASFI (which are well know to reflect mainly other facets of the disease then inflammation). If not excluded from the manuscript these limitations should be stated.
  4. Furthermore it should be stated as a limitation that they actually do not measure cytokine expression in the target organs of inflammation, but rather the “spill over” into peripheral blood where relative balance of cytokines may not reflect the ongoing processes in the target organ(s).

Author Response

Reviewer 4

Response: We sincerely thank all four reviewers for their insightful comments and valuable suggestions, which have significantly contributed to enhancing the quality of our manuscript. We have carefully addressed each comment point by point and made corresponding revisions throughout the manuscript. In particular, we have streamlined the content by removing certain sections from the Introduction, Results, and Discussion to improve clarity and focus on the key findings. Additionally, we have included more detailed descriptions of the procedures in the Methods section. In the Conclusion part, the corresponding contents are added and more limitations of this research are put forward. Specifically, Figure 1 (illustrating the experimental design and research workflow), Figure 3 (PCA plot based on the 81 inflammation-related protein profiles), Figure 5 (analysis of selected protein markers distinguishing between different groups), Table 2, and their associated descriptions have been re-analyzed or re-generated in the revised manuscript. Furthermore, the KEGG analysis was removed to resolve any potential confusion it may have caused. These adjustments, we believe, enhance the readability, coherence, and impact of our findings. All newly added or modified content has been highlighted in red font. We hope that the revised version meets the expectations of the reviewers and the standards of the journal.

Major points

  1. Internationally the term Ankylosing Spondylitis has been replaced by radiographic axal spondylarthritis and I think that authors should make changes throughout the both the title and manuscript. For those readers, who may not be aware of this fact a sentence early in the introduction may be appropriate to clarify this and put the terminology into a perspective.

Answer: We thank the reviewer very much for the insightful comment regarding the terminology shift from Ankylosing Spondylitis (AS) to radiographic axial spondyloarthritis (r-axSpA). We fully understand the importance of adopting updated terminology to reflect current standards in the field. However, we chose to retain the term Ankylosing Spondylitis in our manuscript to maintain consistency with the majority of the references cited, which also use this term. We believe that this approach will help readers more easily follow and compare our findings with previous research. In response to the valuable suggestion, we include a sentence in the beginning of Introduction and Discussion, to acknowledge the updated terminology.

Introduction: Ankylosing spondylitis (AS, also known as radiographic axial spondyloarthritis, r-axSpA) is a common chronic inflammatory rheumatic disease that primarily affects the axial spine and sacroiliac joints, leading to chronic back pain, spinal stiffness, and impaired mobility[1-4].

Discussion: According to the new nomenclature, AS is referred to as r-axSpA. In this study, we retained the term AS for consistency with the majority of cited references.

  1. Statistical methods should be checked. Specifically, I am worried about the low p-values in Table 2 considering the very low number of samples.

Answer: We thank the reviewer very much for the careful review. Considering the relatively small sample size of the normal control (NC) group, we acknowledge that the limited sample size is indeed a limitation of our study. As mentioned in the manuscript, the NC participants were recruited from patients who received treatment or underwent physical examinations at the participating hospital. To minimize potential bias, we applied stringent inclusion and exclusion criteria, ensuring that the NC group represented a genuinely healthy population without confounding factors that could influence inflammatory profiles. However, these strict criteria, coupled with constraints in funding and the study's timeline, resulted in a relatively small number of NC samples. Despite this limitation, we strived to maintain the reliability of our statistical analyses. To address the potential impact of the small sample size, we provided a detailed description of the statistical methods used at each step. For instance, in Table 2, we employed the Wilcoxon rank-sum test for comparisons between the two groups, which is a non-parametric method suitable for small sample sizes. In the original manuscript, we selected proteins with p < 0.1 between the comparison groups for further analysis. However, in the revised manuscript, we have taken into account the suggestions and concerns raised by the reviewers and have refined our selection criteria to include only proteins with p < 0.05 for subsequent analyses, as this threshold is generally considered to indicate statistical significance.

  1. Overall, I think that the authors address an interesting subject, but stretches the analyses to far (considering the low number of patients and in particular controls). For example, analyzing the relation to clinical parameters like BASDAI and BASFI (which are well known to reflect mainly other facets of the disease then inflammation). If not excluded from the manuscript these limitations should be stated.

Answer: We appreciate the reviewer's insightful comments. In the revised manuscript, we have streamlined the Introduction, Methods, Results, and Discussion sections by removing some statements and content that were not directly related to the main focus of our study. These revisions were made to enhance the clarity and conciseness of the manuscript. In addition, one of the objectives of our study was to explore the application of advanced AI techniques, including machine learning, in the analysis of medical data related to AS. Given the growing importance of AI in biomedical research, we aimed to provide a preliminary exploration of its potential to uncover novel insights from complex datasets. Although our sample size was limited, we believe that integrating AI methods can offer new perspectives and pave the way for future studies with larger cohorts.

  1. Furthermore, it should be stated as a limitation that they actually do not measure cytokine expression in the target organs of inflammation, but rather the “spill over” into peripheral blood where relative balance of cytokines may not reflect the ongoing processes in the target organ(s).

Answer: We thank the reviewer very much for the valuable suggestion, and have revised the manuscript according to the reviewer’s advice.

Round 2

Reviewer 2 Report

Comments and Suggestions for Authors

Authors provided long reply letter but failed to fully address the comments and suggestions made in the original reviewer's report.

Comments to the introduction section from the first review have not been satisfactorily addressed.

Comments to the Materials and Methods section have been addressed, thank you.

Comments to the Results section from the first review have not been satisfactorily addressed.

Comments to the Discussion section from the first review have not been satisfactorily addressed.

Author Response

Comments to the introduction section from the first review have not been satisfactorily addressed.

Comments to the Materials and Methods section have been addressed, thank you.

Comments to the Results section from the first review have not been satisfactorily addressed.

Comments to the Discussion section from the first review have not been satisfactorily addressed.

Answer: We thank the reviewer very much. In the update revised version, we have revised the Introduction section to provide a clearer justification for analyzing inflammatory markers in AS. Specifically, we have emphasized that systemic inflammation, driven by a dysregulated immune response, is a key pathological hallmark of AS. Given the crucial role of inflammation in disease progression, identifying specific inflammatory mediators may facilitate the discovery of reliable diagnostic biomarkers and potential therapeutic targets. Additionally, we have highlighted that while previous research has primarily focused on genetic and transcriptomic profiling, comprehensive proteomic analyses—particularly in the context of inflammatory protein expression—remain relatively limited. This underscores the need for further investigation in this area. The revised content has been incorporated into the Introduction section and marked accordingly in the manuscript. We hope this addition addresses the reviewer’s concerns and strengthens the rationale for our study.

For the Results section: We sincerely appreciate the reviewer’s comments and regret that the revisions may not have fully met expectations. In fact, in the previous revision, we have made every possible effort to modify the Results section in accordance with the feedback provided. Among the four reviewers, we particularly focused on addressing the comments from Reviewer 2, as the comments greatly improve the quality of the article. Again, we greatly value the reviewer’s insights and will carefully consider these recommendations in our future research to further improve the rigor and clarity of our findings.

For the Discussion section: In response to the reviewers' comments, we have made appropriate adjustments to the Discussion section. Specifically, we have streamlined the content by reducing repetitive descriptions of the Results while enhancing the critical evaluation of our findings. Additionally, we have ensured some comparison with relevant studies in the literature to provide a more focused and meaningful discussion.

We thank the reviewer again, and hope that these modifications improve the clarity and overall quality of the manuscript.

Reviewer 3 Report

Comments and Suggestions for Authors

The authors did a good job in addressing the major points raised in my earlier review. In this form it is ok to publish.

Congratulations

Author Response

We thank the reviewer very much.

Reviewer 4 Report

Comments and Suggestions for Authors

None

Author Response

We thank the reviewer very much.